# Robot-Assisted Radical Prostatectomy Performed with the Novel Surgical Robotic Platform Hugo™ RAS: Monocentric First Series of 132 Cases Reporting Surgical, and Early Functional and Oncological Outcomes at a Tertiary Referral Robotic Center

**DOI:** 10.3390/cancers16081602

**Published:** 2024-04-22

**Authors:** Angelo Totaro, Eros Scarciglia, Filippo Marino, Marco Campetella, Carlo Gandi, Mauro Ragonese, Riccardo Bientinesi, Giuseppe Palermo, Francesco Pio Bizzarri, Antonio Cretì, Simona Presutti, Andrea Russo, Paola Aceto, Pierfrancesco Bassi, Francesco Pierconti, Marco Racioppi, Emilio Sacco

**Affiliations:** 1Department of Urology, Agostino Gemelli Hospital Foundation—IRCCS, Catholic University Medical School, 00167 Rome, Italy; 2Department of Urology, Isola Tiberina—Gemelli Isola Hospital, Catholic University Medical School, 00167 Rome, Italy; 3Department of Anesthesia, Emergency and Intensive Care Medicine, Agostino Gemelli Hospital Foundation—IRCCS, Catholic University Medical School, 00167 Rome, Italy; 4Department of Pathological Anatomy, Agostino Gemelli Hospital Foundation—IRCCS, Catholic University Medical School, 00167 Rome, Italy

**Keywords:** intraoperative technical data, Medtronic Hugo^TM^ RAS system, prostate cancer, robot-assisted radical prostatectomy, robotic surgery, surgical outcomes

## Abstract

**Simple Summary:**

New devices are being developed and proliferated worldwide to perform robotic surgery. The sharing of data resulting from performing a large number of procedures with the Hugo^TM^ RAS system aims to demonstrate its reliability and potential use in different scenarios. In our case series, this new device has proven to be reliable in performing robot-assisted radical prostatectomy. We list the solutions we applied to address the technical issues of the system and show that these did not have a significant impact on patients and procedures.

**Abstract:**

Background: Robotic-assisted surgery is the gold standard for performing radical prostatectomy (RARP), with new robotic devices such as Hugo^TM^ RAS gaining prominence worldwide. Objective: We report the surgical, perioperative, and early postoperative outcomes of RARP using Hugo^TM^ RAS. Design, setting, and participants: Between April 2022 and October 2023, we performed 132 procedures using the Montsouris technique with a four-robotic-arm configuration in patients with biopsy-proven prostate cancer (PCa). Outcome measures: We collected intraoperative and perioperative data during hospitalization, along with follow-up data at predefined postoperative intervals of 3 and 6 months. Results and limitations: Lymphadenectomy was performed in 25 procedures, with a bilateral nerve-sparing technique in 33 and a monolateral nerve-sparing technique in 33 cases. The mean total surgery time was 242 (±57) min, the mean console time was 124 (±48) min, and the mean docking time was 10 (±2) min. We identified 17 system errors related to robotic arm failures, 9 robotic instrument breakdowns, and 8 significant conflicts between robotic arms. One post-operative complication was classified as Clavien–Dindo 3b. None of the adverse events, whether singular or combined, increased the operative time. Positive margins (pR1) were found in 54 (40.9%) histological specimens, 37 (28.0%) of which were clinically significant. At 3 and 6 months post-surgery, the PSA levels were undetectable in 94.6% and 92.1% of patients, respectively. Social urinary continence was regained in 86% after 6 months. Limitations of our study include its observational monocentric case-series design and the short follow-up data for functional and oncological outcomes. Conclusions: Our initial experience highlights the reliability of the Hugo^TM^ RAS system in performing RARP. Additionally, we also list problems and solutions found in our daily work.

## 1. Introduction

Robot-assisted surgery is currently being used in various surgical disciplines around the world. The daVinci platform from Intuitive Surgical Inc. (Sunnyvale, CA, USA) has been the dominant platform in robotic surgery for the past 20 years [1,2]. Following the expiry of Intuitive’s patent, several alternatives are now emerging [3,4].

The Hugo™ Robotic Assisted Surgery (RAS) system (Medtronic, Minneapolis, MN, USA) is emerging as an innovative and cost-effective solution. This new system has received European CE approval for urological, gynecological, and general surgical procedures in adults [5]. The system is a modular multi-cart robotic platform for robot-assisted minimally invasive surgery. The main components of the system are the surgeon’s console, the system tower, and the arm carts. This platform enables the surgeon, who sits at an ergonomically adjustable remote console, to visualize the surgical field in three dimensions using special glasses. As with other robotic platforms, the surgeon can use two pistol-like controllers on the console to move the endoscope and instruments on the operating table.

For decades, robotic-assisted laparoscopic radical prostatectomy (RARP) has been considered the gold standard in the treatment of localized prostate cancer (PCa). Data on patients who have undergone RARP with the Hugo^TM^ robotic platform are still limited. There is a lack of information on follow-up and potential technical problems.

As a urology center that has pioneered the introduction of this new robotic surgical system, we wanted to report on the perioperative and early postoperative outcomes of one of the largest series of RARP procedures using the Hugo^TM^ RAS system.

## 2. Materials and Methods

A prospective clinical study was planned to evaluate the performance of the Hugo^TM^ RAS system at our center (Agostino Gemelli Hospital Foundation—IRCCS, Catholic University Medical School, Rome, Italy), the first to use this new robotic platform in Italy, and to analyze the surgical and perioperative outcomes of patients who underwent RARP with a first short follow-up.

Before starting the study, approval was obtained from our institution’s ethics committee (ID 5119/2022), and informed consent was acquired from patients prior to surgery. Patients provided consent for the collection of intra- and post-operative data, and no decision deviating from standard clinical practice and guidelines were made. Furthermore, no specific rationale was identified for selecting this robotic platform over the daVinci robotic system, which has been established in our center for several years. The entire surgical team underwent formal technical training online and on-site.

The study enrolled consecutive patients with biopsy-proven pelvic-confined prostate cancer (Pca) for which curative surgical treatment was indicated who underwent RARP and optional pelvic lymph node dissection (PLND) with the Hugo^TM^ RAS system between April 2022 and October 2023.

Preoperative multiparametric magnetic resonance imaging (mpMRI) of the prostate was performed in all patients.

Preoperative staging was performed in patients classified as high or very high risk or in the highest intermediate risk subgroup according to the EAU risk groups for biochemical recurrence of localized and locally advanced prostate cancer. Staging was performed using whole-body CT scan plus bone scintigraphy or PET-CT with choline.

Exclusion criteria included refusal to sign the informed consent form and missing data. However, patients who had previous major abdominal surgery and those who received pelvic radiotherapy or trans-urethral resection of the prostate were excluded from the initial series of ten cases to allow surgeons a gradual transition to the new robotic platform by starting with simpler cases.

Four surgeons with extensive experience with the daVinci platform (more than 300 cases) and two surgeons who were still in the learning phase performed the procedures.

A four-robot-arm configuration was used. RARP was performed according to the Montsouris transperitoneal technique. PLND was performed in patients with a preoperative risk of nodal involvement of more than 7% according to the 2018 version of the Briganti nomogram [6,7]. Depending on the risk of ipsilateral extracapsular extension (based on cT stage, ISUP grade, magnetic resonance imaging, and use of Partin tables), a nerve-sparing procedure was performed.

If necessary, a “fish-mouth” reconstruction of the bladder neck with detached points was performed.

The intraoperative data included robotic issues such as the following:-Red system errors: at least one of the robotic arms not responding to the console, requiring shutdown and restart.-Yellow errors: blocked robotic arms that required removal and replacement of the arm with or without the trocar.-Significant conflicts between the robotic arms: when their positions interfere with each other’s movement.-Broken instruments: the laparoscopic robotic instrument was damaged and had to be replaced.

All perioperative data were collected prospectively.

Postoperative data were collected until January 2024 by a researcher not involved in the surgery during follow-up visits or by telephone interviews at predefined postoperative time points of 3 and 6 months. We also reported data on the following:-Social continence rate: defined as the use of no more than one pad per day [8].-Unfavorable positive surgical margins: a single positive margin greater than or equal to 3 mm or a multifocal positive margin [9].-Erectile function: using the International Index of Erectile Function Questionnaire (IIEF-5).

Our internal protocol provides the removal of the urinary catheter for all patients after 15 days, contingent upon the completion of a cystography confirming the absence of urinary extravasation. During the three months post-surgery, patients are instructed to perform daily pelvic floor muscle exercises, with the option of transitioning to electrical stimulation of the pelvic floor muscle if urinary incontinence persists after a minimum of six months. Additionally, all patients are prescribed Phosphodiesterase-5 inhibitors therapy to be taken three times a week, unless contraindicated, and an intracavernous injection of prostaglandin E1 may be considered after at least six months if erectile dysfunction persists.

Statistical analysis and reporting:

A prospective Microsoft Excel database was used to collect all data. Continuous data were reported using mean and standard deviation (SD). Otherwise, the median and interquartile range (IQR) were used. For prevalence data, the number of observations was expressed as a percentage of the total. The median test for independent samples was used to compare partial and total operative times. Statistical significance was set at a *p*-value < 0.05. All statistical analyses were performed using IBM SPSS Statistics for Windows software, version 26.0 (2019, Armonk, NY, USA: IBM Corp).

## 3. Results

### 3.1. Pre-Operative Data

One hundred and thirty-two patients were enrolled.

The initial data of the patients are listed in Table 1.

### 3.2. Intra-Operative Data

The first 85 procedures were performed only by experienced surgeons who had already completed their learning curve for robot-assisted surgery with the daVinci system in previous years [10].

The subsequent 47 procedures were performed for at least one-third of the operating time by two other surgeons who were at different stages of the robotic surgery learning curve.

Table 2 shows the intra-operative data. Prior to surgery, five patients had radiologic suspicion of pelvic lymph node invasion (cN+) and twenty-five patients underwent PLND. The estimated median blood loss was 100 mL (IQR 100), with no patient requiring intraoperative transfusion.

During the procedures, we noted 12 yellow and 5 red errors on the robotic platform.

In one case, we had to restart the entire system, which led to a significant delay in operation times (total operation time of 374 min). Yellow errors were resolved in two cases by removing and replacing the arm and port, while, in the remaining eight cases, they were resolved by removing the instrument alone was sufficient to resolve the error. Most errors (8 out of a total of 17) occurred during the first nine procedures, as shown in Figure 1. Between interventions number nine and ten, a major software update was performed, which led to a significant decrease in the frequency of red and yellow errors (Figure 1).

During surgery, there were eight cases of significant conflict between instruments and nine cases of broken instruments (four cases involving Maryland bipolar forceps and five cases the monopolar scissors). The broken instruments were replaced immediately without any consequences for the patients or the procedures themselves.

There were no significant intraoperative complications.

As shown in Table 3, no statistically significant differences in console and total operation times were found between the cases with and without the occurrence of errors (cumulative and separately categorized as yellow or red), conflicts, broken instruments, and general technical robotic issues (considered as presence of at least one of the aforementioned events, 34 cases in total).

### 3.3. Post-Operative Data

The postoperative data are shown in Table 4.

Our internal protocol for postoperative pain management prioritizes minimizing opioid usage to prevent delayed canalization, nausea, and vomiting. Pain relief therapy is tailored to individual needs, with a maximum dosage of 1 g of paracetamol three times daily, or the administration of non-steroidal anti-inflammatory drugs (NSAIDs) and opioids if pain remains unresponsive, ensuring patient safety by considering any known allergies. Notably, only 0.7% of patients utilized opioids, while 49% required paracetamol for pain management.

During hospitalization, we observed the following complications: three patients with transient fever on the first postoperative day, which was resolved with empirical antibiotic therapy (Clavien–Dindo I); one patient with transient acute renal insufficiency (increase in serum creatinine up to 3 mg/dL), without signs of hydronephrosis, which was resolved with intravenous hydration (Clavien–Dindo I); one patient with reactivation of Crohn’s disease, which was treated pharmacologically (Clavien–Dindo I); a patient with acute bleeding from the abdominal wall after removal of the drainage, which was stopped with mechanical compression (Clavien–Dindo II); a patient with postoperative lymphocele that required temporary positioning of a percutaneous drainage (Clavien–Dindo IIIa); and a patient who required a second abdominal operation due to jejunal perforation caused during the lysis of adhesions due to a previous gastrojejunostomy (Clavien–Dindo IIIb).

## 4. Discussion

In recent years, following the expiration of Intuitive’s patent, several robotic platforms have emerged in the market. The introduction of the Hugo^TM^ RAS system has sparked considerable curiosity, given its focus on wristed instruments, improved ergonomics, and enhanced 3D imaging. However, there remains a dearth of robust literature detailing the performance and outcome of this new platform, particularly in urological procedures.

In this context, as the second European center to use Hugo^TM^ RAS chronologically, we aimed to present the initial clinical experience and perioperative and early postoperative outcomes of 132 patients who underwent RARP with the Hugo^TM^ RAS system. We previously described our surgical setup with this platform [11]. As this is a case series, we did not select our patients. Among them, there are cases with preoperative ISUP grade 5 (5 cases), patients with severe obesity (BMI > 30, 15 cases), patients with previous major abdominal surgery (21 cases) or previous prostate adenomectomy (4), and patients with high total prostate volume (>100 mL, 6 cases).

The feasibility of performing RARP with the Hugo^TM^ RAS system was evaluated for the first time on cadavers to test the setup of the robotic platform [12]. Subsequently, several case series evaluating the initial performance of Hugo^TM^ RAS system in RARP have been published [13,14,15,16,17,18]. Among these, the largest series is from Bravi et al. [13], who described the surgical outcomes of the first 112 patients treated at a high-volume center. They reported optimal perioperative outcomes with relevant data on early oncologic and functional outcomes.

Our study is the first to report that a comprehensive analysis of technical errors occurred during the surgical procedure. Notably, most of the technical problems occurred within the initial 10 procedures, regarding hardware and software functioning. Following a software update developed by the technical department, the frequency of errors significantly decreased. It is noteworthy that these errors did not have a significant negative impact on procedure duration, as demonstrated in Table 3. The yellow and red errors were related to the robot’s blocked arms, which, in some cases, led to a temporary interruption of the procedure. To resolve this issue, we had to remove and reposition the instrument or robotic arm. Importantly, these errors did not hinder the continuation of the surgical procedure or the results in any intraoperative complications or conversion to open surgery.

Our results on operative time (as total and partial time) align closely with the literature findings from previous case series [13,14,15]. Initially, our main concern was the reproducibility and standardization of the docking process, which we found to be laborious and time-consuming. However, with subsequent procedures, we observed that the docking process of the independent arm carts became easier and faster as we identified the optimal layout for the arm carts. It is important to note that all surgeons involved in our study received a specific dry- and wet-lab training sessions on the Hugo^TM^ RAS docking system and console controls at ORSI Academy (Melle, Belgium). Additionally, team-based training sessions, involving bedside assistants and scrub nurses, were conducted to ensure uniformity in port placement and arm configuration sequence, thereby enhancing troubleshooting skills. Furthermore, a Medtronic technician in the operating room assists in the initial skill acquisition process, enabling a quick transition to autonomy in docking and reducing the time.

No intraoperative complications were recorded. We observed only one Clavien–Dindo IIIb complication, with the patient requiring a second abdominal surgery. The rate of postoperative Clavien–Dindo ≥ 2 complications was 4%, which is consistent with those reported in other case series using Hugo^TM^ [13,14] and even slightly lower than the rate of 7.6% reported by Bertolo et al. in a recent metanalysis regarding RARP performed with the daVinci platform [19].

Regarding oncological outcomes, the rate of positive surgical margins (40.9%) markedly deviates from the literature, where prevalence rates range between 26% and 32% [19,20,21,22], as well as our prior experience with the daVinci system [10]. However, the rate of clinically significant positive surgical margins (28%) aligns with these findings. In this series, we identified the posterolateral basal region as the primary site for positive surgical margins (25 out of 54). This site experiences the most traction during the development of the posterior plane and isolation of the vascular pedicle. Furthermore, a pathological analysis of cases with wide positive margins highlighted “extensive artifactual phenomena from traction” at the site area. We do not attribute our higher rate of positive surgical margins to surgeon experience or procedural complexity. Rather, as explained above regarding traction phenomena, we believe that a major limitation of the Hugo^TM^ RAS system is the absence of a robotic instrument for “gentle” grasping and traction.

In our case series, the rate of undetectable PSA (<0.1 ng/mL) at 3 months after surgery was 94.6%. Among the 54 patients with positive surgical margins, 4 had detectable PSA after 3 months, all of whom were classified as very high risk with locally advanced disease upon final histological examination. Six months after surgery, the rate of undetectable PSA in a sample of 114 patients was 92.1%. However, the evaluation of postoperative oncological outcomes was limited by the brief duration of the follow-up. Additionally, it is worth noting that this limitation prevents direct comparison with other literature findings, and a more extended follow-up period would be ideal for a comprehensive assessment.

Regarding functional outcomes, the rate of social continence was 75.7% at 3 months and 86% at 6 months after surgery. Our rates appear to be consistent with those reported in the literature regarding RARP performed with the daVinci platform [23,24,25].

Post-operative erections were recorded in 28.5% of patients, with a median IIEF-5 score of 10 at 6 months after surgery, which is consistent with findings reported in the literature [26].

Four surgeons involved in the study have significant experience with the daVinci platform and have achieved proficiency in controlling positive surgical margins with the daVinci platform, as demonstrated in this previous study [10]. This indicates that, in their learning curves, they have reached a turning point or peak point after which the positive surgical margin rate begins to decrease and then reaches a proficiency level without significantly increasing again. However, we noted that operating with the Hugo^TM^ RAS system required a higher mental workload throughout the surgical procedure, despite becoming more familiar with the system over time. The two younger surgeons, who are still in the process of completing their learning curve on positive surgical margins with daVinci, have not noted significant differences. However, a greater number of cases are required to properly assess and compare the learning curves Hugo and daVinci platforms.

Nevertheless, Medtronic offers an artificial intelligence (AI)-powered video management and analytics platform tailored to operating rooms. This platform streamlines the recording, analysis, and dissemination of surgery videos directly via a mobile app for the operator. Using AI algorithms, the system automatically segments videos into procedural steps, presenting a novel tool to enhance the performance of experienced surgeons and facilitate the training of younger surgeons.

In other urological robotic interventions as well, initial reports on the utilization of Hugo^TM^ RAS system seem to validate the feasibility and safety of the procedures conducted [27,28,29,30].

To the best of our knowledge, this is the first study to report intraoperative technical data from a series of consecutive cases. It was interesting and noteworthy to present it in the context of the increasing and wider use of the Hugo^TM^ RAS device.

The dissemination of this new robotic platform may expand the market and trigger competition for the development of a more efficient and less expensive platform, which also offers the opportunity to improve access to robotic surgery.

The study exhibits typical limitations inherent to a monocentric observational design: a small sample size, absence of control group, and the solely descriptive data. Additionally, the relatively short follow-up after surgery represents another limitation. Therefore, future investigations should aim to validate our results, particularly concerning functional and oncological outcomes, over a more extended follow-up period. As a result, generalizing these results to a broader population or reaching definitive conclusions became challenging.

## 5. Conclusions

The presented case series underscores the safety and reproducibility of RARP performed with the Hugo^TM^ RAS system. Further investigation into the use of the new robotic traction instruments is necessary to ascertain the underlying cause of the elevated rate of positive surgical margins. Continuous advancements in technology, updates to the system, and customized training initiatives seem essential to address the initial challenges and unleash the full potential of the Hugo^TM^ RAS system in enhancing surgical outcomes.

Further prospective and randomized studies are required to assess the non-inferiority of this robotic platform in terms of oncological and functional outcomes.

## Figures and Tables

**Figure 1 cancers-16-01602-f001:**
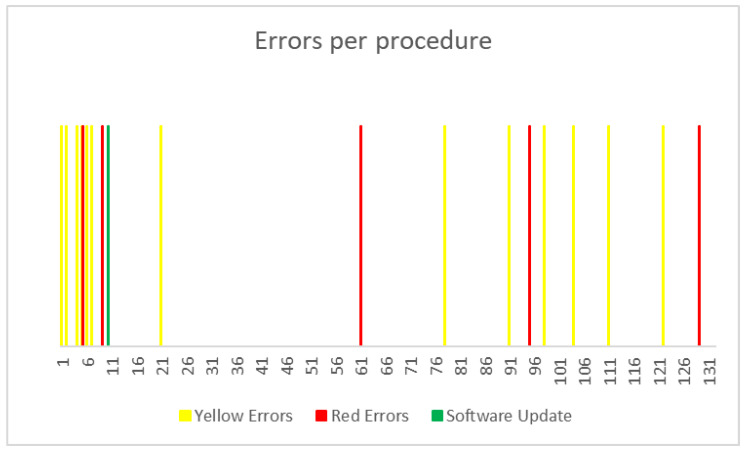
Report on yellow and red errors among procedures in chronological order.

**Table 1 cancers-16-01602-t001:** Baseline patient characteristics.

Age, years, mean (±SD)	66.1 (±6.8)
BMI, kg/mq, mean (±SD)	26.4 (±3.5)
CCI, median (IQR)	5 (1)
ASA score, median (IQR)	2 (0)
Previous abdominal surgery, *n* (%)	61 (46)
- Minor surgery, *n* (%)	36 (59.1)
- Major surgery, *n* (%)	21 (34.4)
- Prostate adenomectomy, *n* (%)	4 (6.5)
IPSS, median (IQR)	7 (9)
QoL, median (IQR)	2 (2)
IIEF-5, median (IQR)	18 (9)
PI-RADS index, median (IQR)	4 (1)
Lesion diameter, mm, mean (±SD)	12.8 (±6.1)
Preoperative PSA level, ng/mL, mean (±SD)	10.72 (±10.81)
Positive digital rectal examination, *n* (%)	27 (20.4)
Prostate volume, mL, mean (±SD)	50.85 (±22.1)
ISUP 1-2 at biopsy, *n* (%)	91 (68.9)
ISUP 3-5 at biopsy, *n* (%)	41 (31.1)
cN+, *n* (%)	5 (3.8)

BMI, Body Mass Index; CCI, Charlson Comorbidity Index; ASA, American Society of Anesthesiologists; IPSS, International Prostatic Symptoms Score; QoL, Quality of Life; IIEF-5, International Index of Erectile Function Questionnaire; PI-RADS, Prostate Imaging-Reporting and Data System; PSA, prostate specific antigen; ISUP, International Society of Urological Pathology; cN+, clinical lymph node involvement.

**Table 2 cancers-16-01602-t002:** Intra-operative data.

Pelvic lymphadenectomy, *n.* (%)	25 (18.9)
Nerve-sparing procedure, total, *n.* (%)	66 (50)
- Bilateral, *n.* (%)	33 (50)
- Monolateral, *n.* (%)	33 (50)
Blood loss, mL, median (IQR)	100 (100)
Intra-operative complications, *n.* (%)	0 (0)
Red errors, *n.* (%)	5 (4)
Yellow errors, *n.* (%)	12 (9)
Significant robotic arms-conflicts, *n.* (%)	8 (6)
Broken robotic instruments, *n.* (%)	9 (6.8)
Bladder neck reconstruction, *n.* (%)	18 (13.6)
Total surgery time (in–out), min, mean (±SD)	242 (±57)
- Pelvic lymphadenectomy, min, mean (±SD)	255 (±56)
- No pelvic lymphadenectomy, min, mean (±SD)	239 (±57)
Operative time (incision to last stich), min, mean (±SD)	189.3 (±57.3)
- Pelvic lymphadenectomy, min, mean (±SD)	200 (±68)
- No pelvic lymphadenectomy, min, mean (±SD)	186 (±53)
Console time, min, mean (±SD)	124 (±48)
Docking time, min, mean (±SD)	10 (±2)
Patience entrance to skin incision, min, mean (±SD)	37.8 (±13.2)
Last stitch to patience exit, min, mean (±SD)	18.8 (±8.1)

**Table 3 cancers-16-01602-t003:** Comparisons of operative times between presence and absence of errors, conflicts, broken instruments, technical robotic problems, and the presence of any technical robotic issues.

Total	Console Time, Min, Median (IQR)		117 (79)	
	Total Surgery Time (in–out), Min, Median (IQR)		232 (77)	
Type of Adverse Event		Absence	Presence	*p*-Value
Errors cumulative (yellow or red)	Console time, min, median (IQR)	112 (80)	135 (87)	0.119
	Total surgery time (in–out), min, median (IQR)	235 (74)	214 (128)	0.603
Yellow error	Console time, min, median (IQR)	115 (79)	120 (76)	0.559
	Total surgery time (in–out), min, median (IQR)	238 (75)	210 (93)	0.243
Red error	Console time, min, median (IQR)	113 (80)	138 (93)	0.068
	Total surgery time (in–out), min, median (IQR)	230 (76)	250 (115)	0.362
Conflicts	Console time, min, median (IQR)	117 (80)	116 (106)	0.715
	Total surgery time (in–out), min, median (IQR)	230 (74)	248 (121)	0.274
Broken instruments	Console time, min, median (IQR)	120 (81)	88 (35)	0.167
	Total surgery time (in–out), min, median (IQR)	215 (83)	219 (46)	0.381
Technical robotic issues	Console time, min, median (IQR)	118 (81)	112 (52)	1.000
	Total surgery time (in–out), min, median (IQR)	240 (78)	210 (65)	0.100

**Table 4 cancers-16-01602-t004:** Postoperative data.

Post-operative pain (VAS in recovery), median (IQR)	0 (1)
Post-operative complication—Clavien–Dindo grade, median (IQR)	1 (1)
- Clavien–Dindo grade I, *n.*	5
- Clavien–Dindo grade II, *n.*	1
- Clavien–Dindo grade III, *n.*	2
Catheter removal (POD), median (IQR)	15 (6)
POD of discharge, median (IQR)	3 (1)
Narcotic use, *n.* (%)	1 (0.7)
NSAIDs use, *n.* (%)	4 (3)
Paracetamol use, *n.* (%)	65 (49.2)
Prostate volume at final pathology, mL, mean (±SD)	44.4 (±19.7)
Tumor volume at final pathology, mL, mean (±SD)	2.77 (±4.5)
Primary Gleason at final pathology, median (IQR)	3 (1)
Secondary Gleason at final pathology, median (IQR)	4 (1)
ISUP at final pathology, median (IQR)	2 (2)
Perineural Invasion at final pathology, *n.* (%)	113 (86.9)
Global percentage of neoplasia, median (IQR)	6 (10)
Positive surgical margins, *n.* (%)	54 (40.9)
- Clinically significative positive surgical margins [9], *n.* (%)	37 (28)
pT stage	
- pT2a, *n.* (%)	2 (1.5)
- pT2c, *n.* (%)	96 (72.7)
- pT3a, *n.* (%)	19 (14.4)
- pT3b, *n.* (%)	15 (11.3)
pN stage	
- pN0, *n.* (%)	21 (15.1)
- pN1, *n.* (%)	4 (3)
- pNx, *n.* (%)	107 (81)
Follow-up data	
- Undetectable PSA (<0.1 ng/mL) at 3 months, *n*. (%)	125 (94.6)
- Undetectable PSA (<0.1 ng/mL) at 6 months ^§^, *n*. (%)	105 (92.1)
- Social continence rate at 3 months, *n.* (%)	100 (75.7)
- Social continence rate at 6 months ^§^, *n.* (%)	98 (86.0)
- IIEF-5 at 3 months, median (IQR)	9 (11)
- IIEF-5 at 6 months ^§^, median (IQR)	10 (12)

VAS, Visual Analogue Scale; POD, post-operative day; NSAIDs, non-steroidal anti-inflammatory drugs; IIEF-5, International Index of Erectile Function Questionnaire. ^§^ Available for 114 patients.

## Data Availability

The data presented in this study are available anonymously upon request from the corresponding author.

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
