# Peer review of "Robot-Assisted Radical Prostatectomy Performed with the Novel Surgical Robotic Platform Hugo™ RAS: Monocentric First Series of 132 Cases Reporting Surgical, and Early Functional and Oncological Outcomes at a Tertiary Referral Robotic Center"

_cancers, 2024, doi:10.3390/cancers16081602_

Round 1

Reviewer 1 Report

Comments and Suggestions for Authors

In the manuscript entitled "Robot assisted radical prostatectomy performed with the novel surgical robotic platform HUGOTM RAS: Monocentric first series of 132 cases reporting surgical, and early functional and oncological outcomes at a tertiary referral robotic center", the authors present a single center's experience using the HUGO RAS robotic surgery system in RALP in a prospective analysis. The development of robotic surgery and the emergence of new systems that have different details and may therefore influence the course of procedures performed justify articles such as the one presented for review. The results presented by the authors are a valuable source for surgeons from centers where the new system will be used.

The work is well written and well structured.

My comments:

 - Operation times are given for all types - I think it would be worth separating procedures with PLND from those without lymphadenectomy.

 - In the discussion, the authors drew attention to the complicated docking process, which needed to be improved so as not to be time-consuming - I think that describing the technical aspects and nuances that sped up this stage would be very valuable for readers.

Author Response

We thank very much the reviewer for his valuable and in-depth revision that allowed us to improve our paper. Thereafter we reported point by point the corrections/changes that have been done in the revised version according to the reviewers’ suggestion.

Reviewer 1
In the manuscript entitled "Robot assisted radical prostatectomy performed with the novel surgical robotic platform HUGOTM RAS: Monocentric first series of 132 cases reporting surgical, and early functional and oncological outcomes at a tertiary referral robotic center", the authors present a single center's experience using the HUGO RAS robotic surgery system in RALP in a prospective analysis. The development of robotic surgery and the emergence of new systems that have different details and may therefore influence the course of procedures performed justify articles such as the one presented for review. The results presented by the authors are a valuable source for surgeons from centers where the new system will be used. The work is well written and well structured.
1. Operation times are given for all types - I think it would be worth separating procedures with PLND from those without lymphadenectomy.
Response. Following the reviewer’s advice, we have included the operative times separately for procedures with and without lymphadenectomy, and we have added a comment in the discussion section.
2. In the discussion, the authors drew attention to the complicated docking process, which needed to be improved so as not to be time-consuming - I think that describing the technical aspects and nuances that sped up this stage would be very valuable for readers.
Response. We agree with the reviewer. We have implemented this aspect in the discussion section, providing our suggestions to sped up this stage.

Reviewer 2 Report

Comments and Suggestions for Authors

1.      Did the study look at long-term urinary and sexual function for more than 3 and 6 months? Knowing how patients do many months or years after surgery helps understand if this robot does as well as others. Comment about it in the discussion.

2.      The article doesn't specify the learning curve comparison directly. How did the learning curve for Hugo™ RAS compare to the Da Vinci system? This would enhance understanding of its ease of adoption. It could be important for hospitals to think about switching or adding robots. Comment about it in the discussion.

3.      The study mentions system errors but lacks a detailed analysis of their error's impact on surgical procedures. A deeper dive into error types, frequency, and resolutions could inform system enhancements. Comment about it in the discussion.

4.      The study presents oncological outcomes with Hugo™ RAS but does not compare them to other systems. This info is vital for assessing the system's effectiveness in cancer treatment. Comment about it in the discussion.

5.      Post-surgery rehabilitation programs are not discussed. Recovery and rehabilitation protocols are beneficial for patient care optimization. Including this information, the reader also understands whether or not there was an influence on the results of the study due to this approach in patients who underwent Hugo™ RAS. Comment about it in the discussion.

6.      The article does not specify if or how missing data were handled. Address this in the methodology to strengthen the study's statistical integrity and ensure robust analysis. This is a key to reliable conclusions.

7.      How does Hugo™ RAS integrate with hospital information systems? Integration with hospital information systems is not covered. Detailing interoperability aspects could clarify how Hugo™ RAS fits into the broader healthcare IT ecosystem. Interoperability is crucial for streamlined workflow.

8.      Could the ethical approval and patient consent process be more detailed? Transparency in these areas is essential for ethical research conduct. Expanding on these processes would enhance transparency and trust in the study's ethical standards. Review the text.

9.      What was the post-operative pain management strategy? Providing details on this aspect would offer the readers a better insight into patient care and recovery after Hugo™ RAS procedures.

10.   The article does not compare complication rates with other studies or systems. Include such a comparison to provide context and aid in evaluating the Hugo™ RAS system's safety profile. Knowing how this system's safety profile compares to others is important for clinical decision-making.

Author Response

We thank very much the reviewer for his valuable and in-depth revision that allowed us to improve our paper. Thereafter we reported point by point the corrections/changes that have been done in the revised version according to the reviewers’ suggestion.
Reviewer 2
1.Did the study look at long-term urinary and sexual function for more than 3 and 6 months?Knowing how patients do many months or years after surgery helps understand if this robot does as well as others. Comment about it in the discussion.
Response. The study only assessed urinary and sexual function outcomes up to 6 months post-surgery. This study aims to provide an initial description of perioperative and early outcomes regarding sexuality, continence, and PSA levels. We have acknowledged this limitation in the discussion section and emphasized the need for a future study with longer-term follow-up data to provide a more thorough assessment, and to better compare the safety and efficacy of this robotic platform with others.
2. The article doesn't specify the learning curve comparison directly. How did the learning curve for Hugo™ RAS compare to the Da Vinci system? This would enhance understanding of its ease of adoption. It could be important for hospitals to think about switching or adding robots. Comment about it in the discussion.
Response. As stated in the Materials and Methods section, we opted to exclude patients who had previous major abdominal surgery, and who received pelvic radiotherapy or trans-urethral resection of the prostate from the initial 10 cases, anticipating a learning curve for RARP performed with HugoTM RAS system. For a meaningful comparison of learning curves, each surgeon in our study must attain a greater number of procedures to identify the turning points. In this regard, we have included the following in the discussion “In our study, we demonstrated that an experienced robotic surgeon could seamlessly transition from using the daVinci platform to the HugoTM RAS system in RARP surgery without experiencing any clinically significant performance decline. Four surgeons involved in the study have significant experience with the daVinci platform and have achieved proficiency in controlling positive surgical margins with the daVinci platform, as demonstrated in this previous study [10]. This indicates that in their learning curves, they have reached a turning point or peak point after which the positive surgical margin rate begins to decrease and then reaches a proficiency level without significantly increasing again. However, we noted that operating with the HugoTM RAS system required a higher mental workload throughout the surgical procedure, despite becoming more familiar with the system over time. The two younger surgeons, who are still in the process of completing their learning curve on positive surgical margins with daVinci, have not noted significant differences. However, a greater number of cases are required to properly assess and compare the learning curves Hugo and daVinci platforms”
3.The study mentions system errors but lacks a detailed analysis of their error's impact on surgical procedures. A deeper dive into error types, frequency, and resolutions could inform system enhancements. Comment about it in the discussion.
Response. The system errors are described in paragraph 3.2. We have added a comment in the discussion regarding their impact on surgical procedures.
4.The study presents oncological outcomes with Hugo™ RAS but does not compare them to other systems. This info is vital for assessing the system's effectiveness in cancer treatment. Comment about it in the discussion.
Response. Regarding oncological outcomes (positive surgical margins and postoperative PSA level) we have provided additional commentary in the discussion section. While it is feasible to compare positive surgical margin rates with findings from other studies conducted using HugoTM RAS system, as well as with other robotic platforms like daVinci, the short duration of follow-up limited the possibility to make comparisons for postoperative PSA level. We advocate for evaluating biochemical recurrence using updated data from longer follow-up period for a more comprehensive analysis. We have added the following: “However, the evaluation of postoperative oncological outcomes was limited by the brief duration of the follow-up. Additionally, it is worth noting that this limitation prevents direct comparison with other literature findings, and a more extended follow-up period would be ideal for a comprehensive assessment”

5.Post-surgery rehabilitation programs are not discussed. Recovery and rehabilitation protocols are beneficial for patient care optimization. Including this information, the reader also understands whether or not there was an influence on the results of the study due to this approach in patients who underwent Hugo™ RAS. Comment about it in the discussion.
Response. We have added a discussion on this important aspect by presenting our internal management protocols for all patients, which has been used for years in our hospital, including procedures performed with daVinci platform. Specifically, we have added the following in the Materials and Methods section: “Our internal protocol provides the removal of the urinary catheter for all patients after 15 days, contingent upon the completion of a cystography confirming the absence of urinary extravasation. During the three months postsurgery, patients are instructed to perform daily pelvic floor muscle exercises, with the option of transitioning to electrical stimulation of the pelvic floor muscle if urinary incontinence persists after a minimum of six months. Additionally, all patients are prescribed Phosphodiesterase-5 inhibitors therapy to be taken three times a week, unless contraindicated, and intracavernous injection of prostaglandin E1 may be considered after at least six months if erectile dysfunction persists.”
6.The article does not specify if or how missing data were handled. Address this in the methodology to strengthen the study's statistical integrity and ensure robust analysis. This is a key to reliable conclusions.
Response. The presence of missing data was an exclusion criterion from enrollment. We added this information to the exclusion criteria described in the Material and Methods section.
7.How does Hugo™ RAS integrate with hospital information systems? Integration with hospital information systems is not covered. Detailing interoperability aspects could clarify how Hugo™™ RAS fits into the broader healthcare IT ecosystem. Interoperability is crucial for streamlined workflow.
Response. The primary focus on interoperability and information integration across computer systems and devices within the new HugoTM RAS platform revolves around a validated mobile app training tool. This application is specifically designed for preparing, practicing, and instructing surgical procedures. We have added the following in the discussion section: “Nevertheless, Medtronic offers an artificial intelligence (AI)- powered video management and analytics platform tailored for operating rooms. This platform streamlines the recording, analysis, and dissemination of surgery videos directly via a mobile app for the operator. Using AI algorithms, the system automatically segments videos into procedural steps, presenting a novel tool to enhance the performance of experienced surgeons and facilitate the training of younger surgeon.”
8.Could the ethical approval and patient consent process be more detailed? Transparency in these areas is essential for ethical research conduct. Expanding on these processes would enhance transparency and trust in the study's ethical standards. Review the text.
Response. The text has been revised to provide additional details regarding the informed consent process.
9.What was the post-operative pain management strategy? Providing details on this aspect would offer the readers a better insight into patient care and recovery after Hugo ™ RAS procedures.
Response. Regarding post-operative management, we have added the following in paragraph 3.3, of the Results section: “Our internal protocol for postoperative pain management prioritizes minimizing opioid usage to prevent delayed canalization, nausea, and vomiting. Pain relief therapy is tailored to individual needs, with a maximum dosage of 1 gram of paracetamol three times daily, or the administration of non-steroidal anti-inflammatory drugs (NSAIDS) and opioids if pain remains unresponsive, ensuring patient safety by considering any known allergies. Notably, only 0.7% of patients utilized opioids, while 49% required paracetamol for pain management.”

10.The article does not compare complication rates with other studies or systems. Include such a comparison to provide context and aid in evaluating the Hugo™ RAS system's safety profile. Knowing how this system's safety profile compares to others is important for clinical decision-making.
Response. We agree with the reviewer and have added a comment and comparison with other studies conducted with HugoTM RAS system as well as other robotic system. We have included the following: “No intraoperative complications were recorded. We observed only one Clavien-Dindo IIIb complication, with the patient requiring a second abdominal surgery. The rate of postoperative Clavien-Dindo ≥2 complications was 4%, which is consistent whit those reported in other case series using HugoTM [13,14] and even slightly lower than the rate of 7.6% reported by Bertolo el al. in a recent metanalysis regarding RARP performed with the daVinci platform [19]”

Round 2

Reviewer 2 Report

Comments and Suggestions for Authors

The statement "In our study, we demonstrated that an experienced robotic surgeon could seamlessly transition from using the daVinci platform to the HugoTM RAS system in RARP surgery without experiencing any clinically significant performance decline" uses language that might overstate the ease of transition and the equivalence of performance between the two systems. Terms like "seamlessly transition" and "experienced robotic surgeon" are subjective and can create an impression of effortless adaptability and superiority of skill that may not reflect the complex reality of surgical practice. Additionally, the phrase "any clinically significant performance decline" minimizes the potential for nuanced differences or challenges that could arise during such transitions.

It's crucial to be cautious with such statements, as history shows that postoperative complications can often be linked to the surgeon's experience rather than the device used. Notable issues with the daVinci system in the past highlight the risks of overconfidence in technology without sufficient emphasis on the learning curve and variability in individual surgical outcomes.

It might be better to change or remove this sentence to avoid giving the wrong idea, pass transparency to the readers and to prevent potential misconceptions. It is important to honestly show the real situation with new surgical tools and how surgeons adapt to them. 

Please informe if the company provides any training for the urologic surgeon on the HugoTM RAS system, or if this is something the surgeon must arrange independently.

I am, however, satisfied with the answers to the other questions.

Author Response

We thank very much the reviewer for his revision that allowed us to improve our paper. Thereafter we reported point by point the corrections/changes that have been done in the revised version according to the reviewers’ suggestion.

Reviewer 2

1.The statement "In our study, we demonstrated that an experienced robotic surgeon could seamlessly transition from using the daVinci platform to the HugoTM RAS system in RARP surgery without experiencing any clinically significant performance decline" uses language that might overstate the ease of transition and the equivalence of performance between the two systems. Terms like "seamlessly transition" and "experienced robotic surgeon" are subjective and can create an impression of effortless adaptability and superiority of skill that may not reflect the complex reality of surgical practice. Additionally, the phrase "any clinically significant performance decline" minimizes the potential for nuanced differences or challenges that could arise during such transitions.

It's crucial to be cautious with such statements, as history shows that postoperative complications can often be linked to the surgeon's experience rather than the device used. Notable issues with the daVinci system in the past highlight the risks of overconfidence in technology without sufficient emphasis on the learning curve and variability in individual surgical outcomes.

It might be better to change or remove this sentence to avoid giving the wrong idea, pass transparency to the readers and to prevent potential misconceptions. It is important to honestly show the real situation with new surgical tools and how surgeons adapt to them.

Response. We agree with the reviewer. The statement regarding the transfer of skills between platforms is somewhat strong and may not accurately reflect reality. Therefore, we have chosen to remove this sentence from the discussion section.

2.Please informe if the company provides any training for the urologic surgeon on the HugoTM RAS system, or if this is something the surgeon must arrange independently.

Response. All members of the surgical team received specialized training at ORSI Academy (Melle, Belgium). Therefore, we have added the following to the discussion section: “It is important to note that all surgeons involved in our study received a specific dry and wet lab training sessions on the HugoTM RAS docking system and console controls at ORSI Academy (Melle, Belgium). Additionally, team-based training sessions, involving bedside assistants and scrub nurses, were conducted to ensure uniformity in port placement and arm configuration sequence, thereby enhancing troubleshooting skills”
